# ETL Best Practices for Data Quality Checks in RIS Databases

**Otmane Azeroual** [1,2,3,*] **, Gunter Saake** [2] **and Mohammad Abuosba** [3]

1   German Center for Higher Education Research and Science Studies (DZHW), Schützenstraße 6a,
    10117 Berlin, Germany
2   Institute for Technical and Business Information Systems—Database Research Group,
    Otto-von-Guericke-University Magdeburg, Universitätsplatz 2, 39106 Magdeburg, Germany;
    Saake@iti.cs.uni-magdeburg.de
3   Department of Computer Science and Engineering, University of Applied Sciences—HTW Berlin,
    Wilhelminenhofstraße 75 A, 12459 Berlin, Germany; Mohammad.abuosba@htw-berlin.de
*   Correspondence: Azeroual@dzhw.eu

**Abstract:** The topic of data integration from external data sources or independent IT-systems has received increasing attention recently in IT departments as well as at management level, in particular concerning data integration in federated database systems. An example of the latter are commercial research information systems (RIS), which regularly import, cleanse, transform and prepare the analysis research information of the institutions of a variety of databases. In addition, all these so-called steps must be provided in a secured quality. As several internal and external data sources are loaded for integration into the RIS, ensuring information quality is becoming increasingly challenging for the research institutions. Before the research information is transferred to a RIS, it must be checked and cleaned up. An important factor for successful or competent data integration is therefore always the data quality. The removal of data errors (such as duplicates and harmonization of the data structure, inconsistent data and outdated data, etc.) are essential tasks of data integration using extract, transform, and load (ETL) processes. Data is extracted from the source systems, transformed and loaded into the RIS. At this point conflicts between different data sources are controlled and solved, as well as data quality issues during data integration are eliminated. Against this background, our paper presents the process of data transformation in the context of RIS which gains an overview of the quality of research information in an institution's internal and external data sources during its integration into RIS. In addition, the question of how to control and improve the quality issues during the integration process in RIS will be addressed.

**Keywords:** research information systems (RIS); heterogeneous information sources; metadata; data integration; data transformation; extraction transformation load (ETL) technology; data quality

## 1. Introduction

In recent years, there has been a new trend in which universities, research institutions and researchers capture, integrate, store and analyze their research information into a research information system (RIS). Other names for the term RIS are current research information system (CRIS), especially in Europe, and RIMS (research information management system), RNS (research networking system) or FAR (faculty activity reporting). The RIS is a central database or federated information system [1,2] that can be used to collect, manage and provide a variety of information about research activities and research results. The information represented here are metadata about an institution's research activities, e.g., personal data, projects, third-party funds, publications, patents and prizes, etc. and

are referred to as research information [3]. The following Figure 1 illustrates the type of research information. Further definitions of the term RIS in the literature can be found in the related papers [4–6].

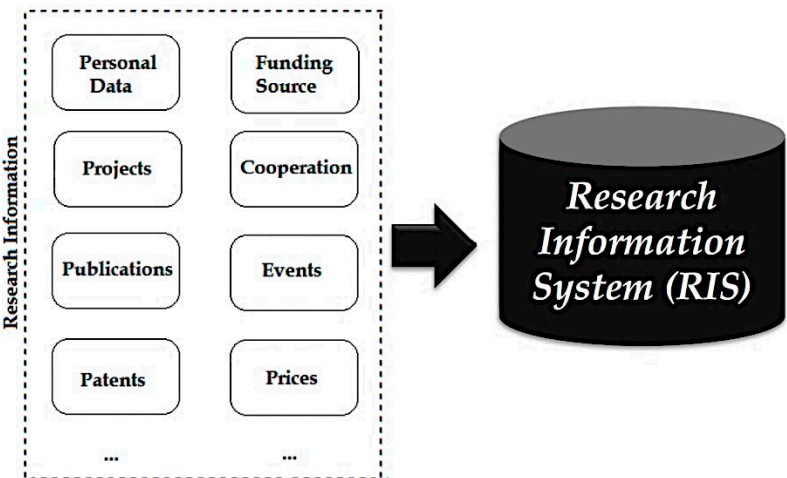

**Figure 1.** Type of research information processed and integrated into research information system.

For the institutions, it is a useful tool to clearly present their own research profile and thus an effective tool for information management. The unification and facilitation of strategic research reporting creates added value for the administrators, as well as for the scientists. The expense of documenting projects is reduced for the scientists, thus gaining more time for the actual research process. The RIS can also support additional helper functions for scientists: e.g., the creation of CVs and publication lists, or other reuse of the aggregated data from RIS on a website.

The most important decision-making and reporting tools in many institutions are RIS that collectively integrate the data from many heterogeneous and independent information systems. In order to avoid structural and content-related data quality issues and thereby reduce the effective use of research information, the transformation of research information takes place during the integration process. "Data integration improves the usability of data by improving consistency, completeness, accessibility, and other dimensions of data quality" [7]. Data integration is the correct, complete and efficient merging of data and content from heterogeneous sources into a consistent and structured set of information for effective interpretation by users and applications [8]. Filling of RIS should already be ensured in good quality as part of the integration process. However, they should be continually defined, measured, analyzed, and improved to increase interoperability with external systems, improve performance and usability, and address data quality issues [9]. Therefore, investing in the topic of data integration is particularly useful when the achievement of high data quality is paramount [10]. This is also the case for RIS users and it is expected that decisions based on relevant, complete, correct and current data will increase the chance of getting better results [11].

The purpose of this paper will be to present the process of data transformation that controls and improves the data quality issues in RIS, to enable institutions to perform in-depth analysis and evaluation on existing or current data, and thus to meet the needs of their decision makers.

The first chapter discusses the causes of data quality issues in RIS that may be responsible for reducing data quality. Subsequently, the phases of the integration process are described in the context of RIS. The chapter concludes with approaches to eliminate data quality issues from external data sources in the transformation phase into RIS.

## 2. Causes of Data Quality Issues in the Context of Integrating Research Information

Data quality defined as "fitness for use" and its assurance is recognized as a valid and important activity, but in practice only a few people list it the highest priority [12]. The quality of the data depends to a great extent on the use of data and synergies of customer needs, the usability and the

access of data [13]. Therefore, during evaluating and improving data quality, the involvement of data users and other data groups is important because they play a high role in data entry, data processing, and data analysis [14]. The subject of data quality should not be considered a one-time action and should therefore be an object of constant care and attention ("quality management"), in order to improve system performance, as well as the user satisfaction and acceptance [2]. Ensuring effective and long-term data quality increasement requires continuous data quality management.

Data integration is a key issue in data quality and is one of the basic activities to improve the quality of data distributed in independent data sources [15]. This is because it can reduce both structural and semantic heterogeneity and redundancy, and increase availability and completeness [15].

Integrating research information with heterogeneous systems into the RIS is a difficult task for the institutions. This is mainly due to the fact that a number of interfaces must be developed in order to allow an exchange of information between the systems [16] and to solve problems related to the provision of interoperability to information systems by the resolution of the heterogeneity between systems on the level of data [17].

Poor or dirty research information designates in a RIS that does not meet the required data quality requirement at a given time. Valid research information (good) becomes invalid research information (bad) if it contains erroneous information, distorting research reporting or even causing serious errors in the RIS. These distortions and errors occur because the research information used contains [2]:

- spelling errors,
- missing values,
- redundant values,
- contradictory values,
- missing data relation,
- different data types,
- different formatting,
- different attributes with the same attribute name,
- different values for the same attributes.

Data quality issues in RIS are generally very diverse, which makes it reasonable to categorize the causes of poor data quality of research information. The causes of data quality issues are in data collection, data transfer, and data integration from the data sources to the RIS. The data sources must be first differentiated into internal and external data sources. Internal data sources are information systems of a research institution that manage internal research information. This includes, in particular, databases of the central administrations (e.g., personnel administration with data on employees, financial accounting with data on projects, income and expenditure and libraries with their data on publications, etc.). External data sources are used in RIS to supplement or replace manual entries. This applies in particular to publications. In many research institutions, scientists have to report their publications. This research information can then be supplemented with external publication databases, and in some cases the research institutions only use external data sources for publication data.

With an increase in various data sources, systems and interfaces in the research management process, the likelihood of data quality issues increases as well. When collecting research information, user errors (such as wrong spelling, missing input, incomplete, incorrect and non-current recording, etc.) are covered. The causes of input errors are a potential source of errors not only for administrators who edit incurred research information accumulated in the institutions. The causer could also be the supporting software systems used, which make false inputs. Reasons for the occurrence of data quality issues are also format differences or conversion errors, etc.

When transferring different heterogeneous systems with different standardized exchange formats (e.g., the Common European Research Information Format [CERIF] data model and the German Research Core Dataset [RCD] data model) in a RIS, new data quality issues may also arise. Particularly problematic is the erroneous mapping of research information from the sources to the RIS [18].

However, the heterogeneity of information systems and their schemas often requires a more expressive mapping language, because the relationships between different data models cannot be captured by simple links with semantics like "sameAs" or "isSimilarTo" [18].

If semantic errors occur, research information can lose all of its informational content, representing a wrong or incomplete part of reality in RIS. This is one of the biggest challenges of integrating research information. Detailed information on these problems and the solution can be found in the related paper [19].

Another common cause of data inconsistency in RIS is that the research information exists in different data models and structures and is collected independently. To define the unified interface, an integrated schema must be designed that covers all aspects of each database and summarizes common aspects [20].

When developing source data systems, there are a variety of error characteristics, e.g., poor database design, in which referential integrity, as well as dependencies and uniqueness, have not received enough attention, e.g., duplicate key IDs are not considered in the case of incorrect modelling, which can lead to massive problems when they are used in a RIS, which can enormously increase the cost of integration or even jeopardize the overall integrity of the data source system [21].

The explained causes of insufficient research information clarify the complexity of data quality issues in the daily work of the RIS. But they are just starting points where such causes can be detected in universities and research institutions and so focus on treatment centers of data quality issues. Where just the elimination of the causes promise better data quality of research information. Universities and research institutions face this huge challenge because the biggest impact of poor data quality is usually the cost of strategic mismanagement by research management due to erroneous data.

## 3. Integration of Research Information—Extraction, Transformation and Load (ETL)

The volume of data in universities and research institutes is constantly growing, and at the same time, the need to analyze a consistent view increases. Analyzes are intended to provide insights into operational research management as well as help with decision-making in strategic planning and control. In a RIS, research information from all the relevant data sources and information systems (this level contains, for example, databases from the administration) of an institution is brought together and kept in a uniform schema. The filling of RIS takes place via the ETL process. A variety of definitions and explanations of this term can be found in [22,23].

During an ETL process, data is first extracted from source systems. In the following transformation phase, data is cleaned up, processed, and transformed into the target schema. Subsequently, during the loading phase, the previously extracted and transformed data are written to the target system, e.g., RIS. This process is the same as transferring enterprise data to the data warehouse. ETL processes generally play an important role in data integration and are used wherever data from different source systems must be transferred to other data storage systems and adapted to new requirements. The research institutions can either program the ETL processes or can be developed and executed with the help of tools. Due to the high complexity of ETL processes, the use of a tool is highly recommended in most cases [24]. The process of the ETL knowledge recognition process from the integrated data sources in the context of RIS is illustrated by the following Figure 2.

The first step of the ETL process is extraction. In this initial step, a portion of the research information from the sources is usually extracted and provided for transformation. The sources can consist of different information systems with different data formats and data structures. Before the actual extraction, the relevant sources and research information must be identified and selected. Different factors play a role in selection. On the one hand, the quality of research information must be taken into account. For external data sources, this can fluctuate and thus not always be guaranteed in the long term. Once the relevant data has been identified, it must be analyzed during data profiling and checked for its technical quality. Detailed information on data profiling in the context of RIS can be found in the related paper [25].

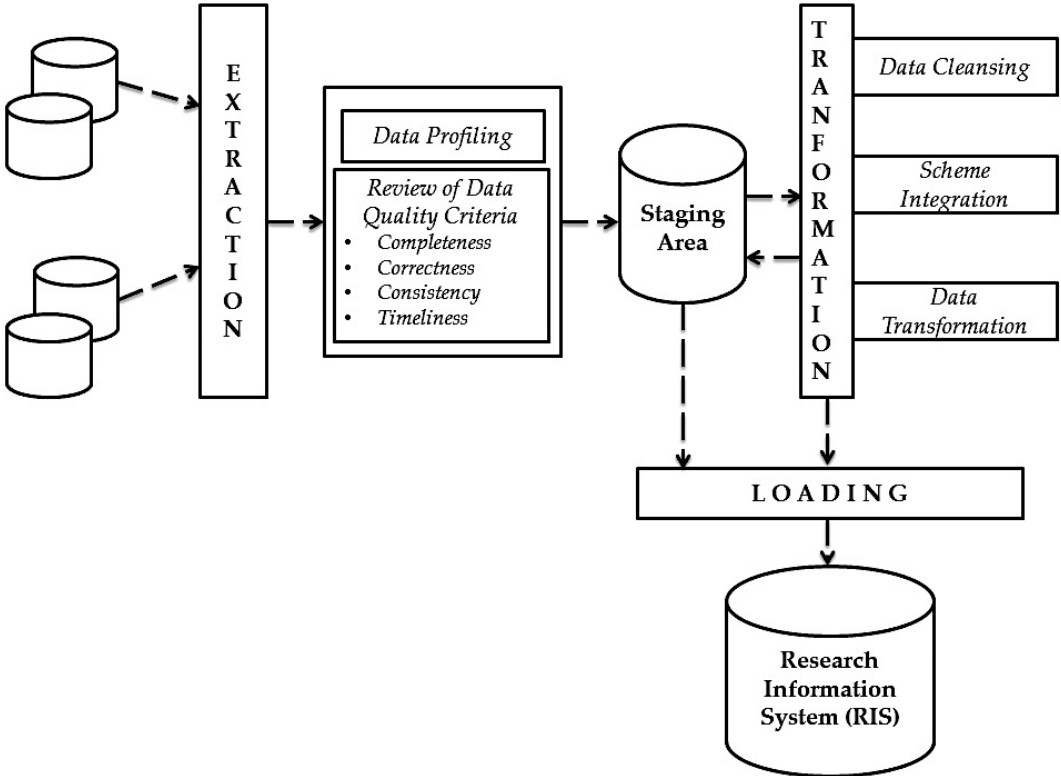

**Figure 2.** Integrating of research information with the help of extract, transform, and load (ETL) process.

Research information from an institution usually needs to be available for different application and usage scenarios. Therefore, it is necessary to define and verify application-dependent quality criteria that can be assessed by appropriate metrics. In the context of RIS, only the four data quality criteria (completeness, correctness, consistency and timeliness) and their corresponding metrics are considered because they are used intensively in scientific institutions to check and measure their data quality. Table 1 shows a quality criteria model for measuring research information in RIS during the ETL process.

The four selected metrics have been found to be easy to measure. In addition, a particularly representative illustration of the reporting for the RIS users and provide an improved basis for decision-making [1]. Detailed information on measuring data quality in RIS can be found in the related paper [1].

These criteria can be used to define requirements for data quality from different stakeholders (in particular from RIS users or RIS administrators) that must be met in order to select the research information. Figure 3 demonstrates these requirements with their possible data quality metrics in the context of RIS, which provide quality criteria determined in the ETL process.

Data quality issues from the operational systems are detected early in the extraction layer to allow early intervention in the time-consuming ETL process. In the requirements for the content correctness, completeness, consistency and timeliness of the data, however, a new calculation of the data quality indicators in the loading layer makes sense. For example, these data quality indicators from the requirement for content correctness are derived, e.g., "share values not in value list", "proportion of inaccurate values" or "proportion of outdated information". These indicators are calculated in the loading layer for data that has been newly created as part of the transformation process. Thus, the substantive correctness of the underlying data can be checked a second time. Furthermore, transformation processes can increase/reduce the amount of data. Therefore, it is important to check if there is enough data after the transformation. In the transformation phase of the ETL process,

measures can be made to improve data quality. There are exactly four measures that can be taken subsequently [24]:

- Pass the entry without error (ignore),
- Pass the entry and mark the affected column (flag),
- Discard/sort out the entry (detect/extract),
- Stop the ETL flow (raise critical error).

**Table 1.** Quality criteria model for measuring the research information during the ETL process.

| | | Data Quality Criteria | Metrics |
|---|---|---|---|
| **Extraction—Transformation and Clean-up—Integration and loading** | Completeness | <ul><li>Relationship between the extracted data and the existing data</li><li>Number of incomplete extraction operations (e.g., aborts due to errors)</li><li>Degree to which the integration and charging process corresponds to the conceptual and logical models</li><li>Relationship between the data actually loaded into the RIS and the expected amount of data</li><li>Number of errors during the process</li></ul> | $Q_{Completeness} = 1 -$ (Number of incomplete units)/(Number of units checked) <br> Degree of achievement = 0–100% |
| | Correctness | <ul><li>The properties of the data must be correct</li><li>The data must be correct in content</li><li>Number of incorrect transformations/clean-up</li><li>Number of crashes due to unexpected errors</li><li>Degree to which the correctness of the data was improved</li></ul> | $Q_{Correctness} = 1 -$ (Number of incorrect in data units)/(Total number of data units) <br> Degree of achievement = 0–100% |
| | Consistency | <ul><li>Number of inconsistent integration or loading instructions</li><li>Number of inconsistent transformations/clean-up</li><li>Number of inconsistent errors during the process</li></ul> | $Q_{Consistency} = 1 -$ (Number of inconsistent units)/(Number of consistency checks performed) <br> Degree of fulfillment= 0–100% |
| | Timeliness | <ul><li>Degree of timeliness, can be reached through the functions</li><li>Number of inaccurate transformations/clean-up</li><li>Number of detected inaccurate data errors</li></ul> | $Q_{Timeliness}(W,A) = e^{(-decline(A).age(W,A)}$ <br> Degree of achievement = 0–100% |

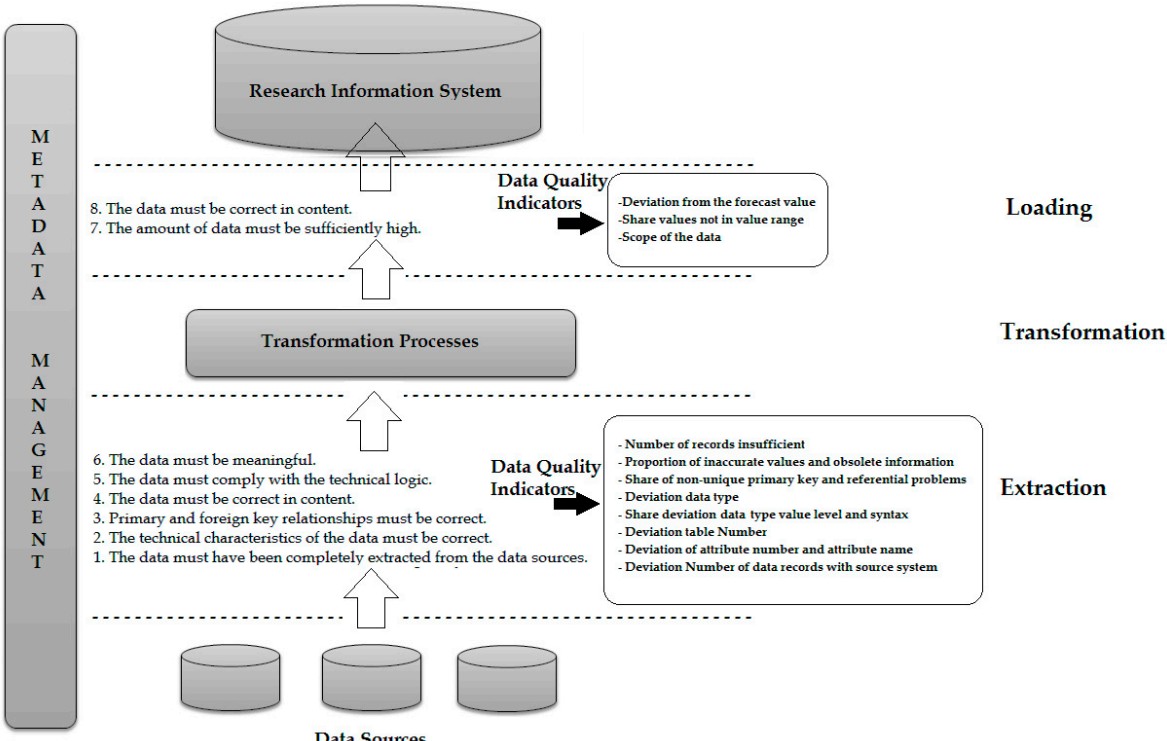

**Figure 3.** Data quality requirement in RIS.

The potential for improving data quality must therefore be considered when selecting research information. After the selection of the research information to be extracted, these are loaded into a so-called staging area. Staging improves the reliability of the ETL process, allowing ETL processes that maintain the research information in a staging area to be rebooted in the event of a failure without re-accessing the data sources [24].

The next step in data integration is the transformation of research information. During the transformation phase, the processing of the research information takes place. This occurs in three sub-phases: data cleansing, schema integration, and data transformation. Detailed information on these phases using the standards of research information (e.g., European CERIF and German RCD) can be found in the related paper [19].

When cleaning up research information, incorrect source data is treated. This is necessary because they could distort the results of the analyzes carried out on the research information. Data cleansing can use a variety of specialized methods (such as parsing, standardization, enhancement, matching, and consolidation) to eliminate errors when integrating multiple data sources. Detailed information on the cleansing of research information can be found in the related paper [3].

Since the data sources come from different operational systems, they are often schematic-heterogeneous. For integration into the RIS, however, these must be converted into a uniform scheme. After the data sources are cleaned up and transformation rules for schema integration are defined, transformations can be defined that adapt and expand the research information according to the requirements of the RIS. After performing all transformations, the loading phase follows. In the final step of the ETL process (loading phase), the adjusted and transformed research information is written to the RIS. Thereafter, the research information can already be used for the RIS analyzes.

## 4. Best Practice in Quality Check during Research Information Transformation

To ensure the quality of research information in the internal and external data sources of an institution during its integration into the RIS. Universities and research institutes have already made numerous attempts to develop solutions to this need. These methods (e.g., cleaning, transforming,

harmonizing and merging) and techniques (e.g., schema mapping and schema integration) of information integration in the context of RIS have been considered and examined in the following paper [19]. So, that it serves as a solution to the problem. In this present paper, we focus on examining data quality issues during the transformation phase of the ETL process because it is an important part of data integration and securing data quality.

Data transformations eliminate the quality issues caused by inconsistent representation or structural conflicts. The whole process is based on firmly defined transformation rules. Solutions for the elimination of possible issues that may arise from internal and particularly external data sources in the transformation phase into the RIS are presented.

Data quality issues during the integration of research information into the RIS can be determined by using column checks. A practical guide in terms of the context of RIS is considered in the following subchapters.

### 4.1. Key Treatment

Keys from external data sources can usually not be transferred to RIS as they must be globally unique. During the transformation, the source keys are mapped to surrogates. The following Table 2 shows an example of mapping from local to global keys.

**Table 2.** Example of mapping from local to global keys.

| Source | Relation | Attribute | Local key | Global Surrogate |
|--------|----------|-----------|-----------|------------------|
| System 1 | Person | Person_Id | 14356 | 55 |
| System 1 | Person | Person_Id | 22222 | 75 |
| System 2 | Publication | Publication_Id | 22222 | 82 |
| System 3 | Project | Project_Id | B469 | 75 |
| System 3 | Project | Project_Id | C101 | 97 |

If global surrogates are identical, they are the same object—in this case the same persons.

### 4.2. Adaption of Data Types

If the data type of a source attribute does not match the corresponding target attribute, then a conversion of the attribute values is required. Table 3 shows the adjustment of data types for an author's date of birth.

**Table 3.** Example of adaptation of data types.

| Character | Number |
|-----------|--------|
| "1989" | 1989 |
| **Character** | **Date** |
| "19.03.1989" | 19.03.1989 |

### 4.3. Conversion of Encodings

A conversion is necessary for attributes that have an encoding, if the encoding standards are different in source and target. However, it may also happen that plain texts are to be encoded or codes must be converted back into plain texts. The following Table 4 demonstrates an example of conversation of encodings based on the gender of the author in a publication.

**Table 4.** Example of conversion of encodings.

| Clear Text | Encoding |
|:---:|:---:|
| Male | 0 |
| Female | 1 |

### 4.4. Unification of Strings and Dates

The strings can be unified by the transformation. An example of the unification of strings is the replacement of umlauts by an alternative spelling. The alternative spelling can only be uppercase or lowercase letters. For the date specification, a distinction is made between internal and external presentation. The internal representation is static and the external is adaptable to the respective user requirements, for example country-specific date format. For this, the example of author name and date of birth is used as follows:

<div align="center">

Müller—*Mueller*

Mueller—*MUELLER*

MM-DD-YYYY—*DD.MM.YYYY*

</div>

### 4.5. Separation and Combination of Attribute Values

Sometimes it may happen that attribute values, which are summarized in the data sources must be split into individual attributes in the target system. This process is called separation, while the opposite case, the merging of individual attributes from the sources, is referred to as a combination of attribute values.

An example of the separation and combination of attribute values based on the date of birth can be illustrated as follows:

<div align="center">

Day= 19, Month = 03, Year = 1989—15.03.1989

</div>

### 4.6. Connecting (Join) and Combining (Union) Data from Multiple Data Sources

Sometimes it can happen that two data sources need to be linked together. In this case, the solution of the node would be of type "join". Table 5 illustrates how the common feature Author_Id can be used to link the two data sources.

Using this transformation, two data sources can be combined. Since, in contrast to the join, only one intersection may be passed on, in the union all fields of the source structure are used in the result. The following Table 6 shows how two data sources are combined and how in contrast to the join, empty attribute values can also be created in the characteristic gender.

### 4.7. Calculation of Derived Values

Sometimes it makes sense to derive new values from certain attribute values of the source. These derived values are then stored in the target source. An example of calculating the derived values' using a funded project is shown as follows:

<div align="center">

Project duration {Start, End} = Funding of project

Funding of project = End of project—Start of project

</div>

**Table 5.** Example for connecting data from multiple data sources.

| Author_Id | DOI | Pub_Year |
|---|---|---|
| 12877 | 10.1002/mop.26174 | 2018 |
| 16738 | 10.1112/S0024611502013692 | 2002 |
| 11545 | 10.1038/nrmicro2496 | 1995 |
| 15336 | 10.1016/j.jhazmat.2011.09.041 | 2012 |

| Author_Id | Title |
|---|---|
| 12877 | Analyzing multimedia streaming in a distributed environment |
| 16738 | Clinical manifestations of diffuse idiopathic skeletal hyperostosis (DISH) |
| 11545 | Early diagnosis of oral cancer using AgNOR-analysis |
| 15336 | Analytic solutions for nonlinear waves in coupled reacting systems |

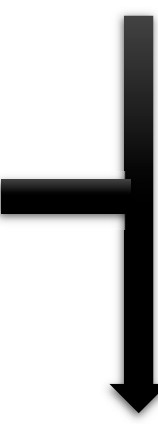

| Author_Id | DOI | Title | Pub_Year |
|---|---|---|---|
| 12877 | 10.1002/mop.26174 | Analyzing multimedia streaming in a distributed environment | 2018 |
| 16738 | 10.1112/S0024611502013692 | Clinical manifestations of diffuse idiopathic skeletal hyperostosis (DISH) | 2002 |
| 11545 | 10.1038/nrmicro2496 | Early diagnosis of oral cancer using AgNOR-analysis | 1995 |
| 15336 | 10.1016/j.jhazmat.2011.09.041 | Analytic solutions for nonlinear waves in coupled reacting systems | 2012 |

**Table 6.** Example for combining data from multiple data sources.

| Author_Id | Birthdate | Gender |
|---|---|---|
| 12877 | 1989-11-17 | M |
| 16738 | 1960-02-20 | M |

| Author_Id | Birthdate |
|---|---|
| 11545 | 1989-11-17 |
| 15336 | 1960-02-20 |

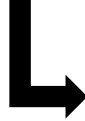
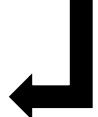

| Author_Id | Birthdate | Gender |
|---|---|---|
| 12877 | 1989-11-17 | M |
| 16738 | 1960-02-20 | M |
| 11545 | 1989-11-17 | |
| 15336 | 1960-02-20 | |

*4.8. Aggregation*

The data can be grouped according to relevant analysis criteria such as age group of researchers, funding of projects or region. The most and simplest way to use aggregation is the summation, e.g., summation of funding a project for a certain period. The following aggregate functions are offered:

- SUM: The values of the aggregation fields and their identical grouping fields are summed.
- MIN: Only the minimum in the aggregation field will be passed.
- MAX: Only the maximum in the aggregation field will be passed on.
- AVG: The average of all values in the aggregation field is calculated and passed on.
- AV0: The average of all values in the aggregation field is calculated, but without considering the null values.
- NOP: No aggregation is performed.

With the help of the transformation of research information during its integration with the ETL process, all errors can be detected. Figure 4 illustrates a practice example of examining and cleansing the publication list from Web of Science and its contents in the transformation phase using a DataCleaner tool.

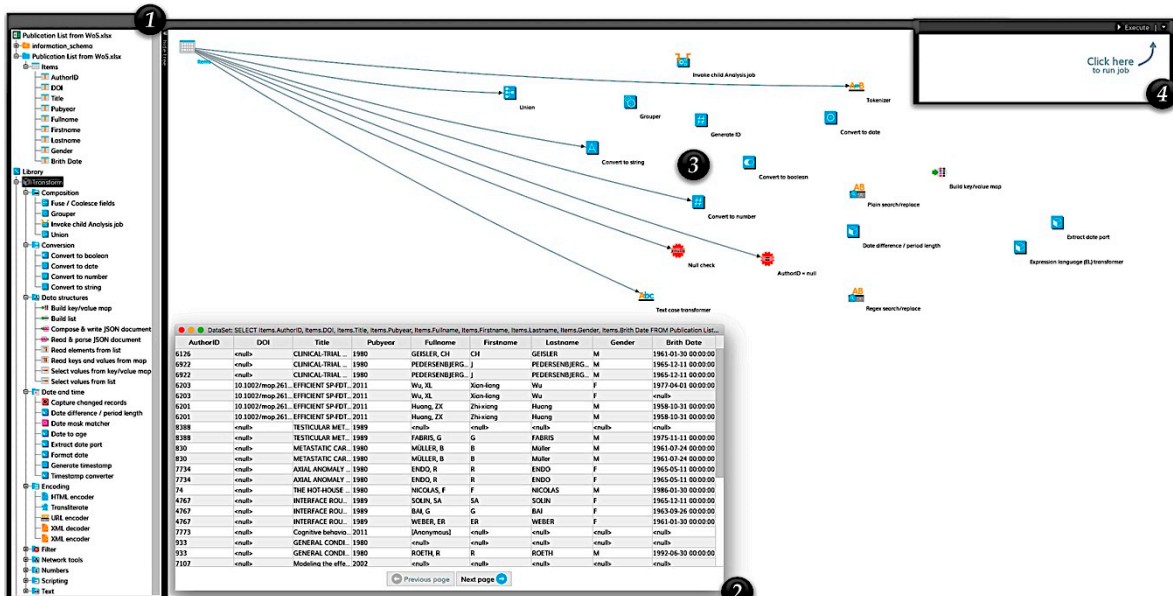

**Figure 4.** Example of a transformation of the publication data using DataCleaner tool.

The *first phase (1)* demonstrates the imported publication data and the components of the transformation (e.g., Composition, Conversion, Data Structures, Data and Time, Encoding, Filter, etc.). DataCleaner provides a variety of transformation components that can be used to process the research information to extract, generate, or refine it. In the *second phase (2)*, the imported publication data is pushed into the system and then opened as a table to check the data again. In the *third phase (3)*, the required components are selected and cabled to the table so that an arrow is created between the components as in the graphic. Click on the right mouse button to select "Link to ..." in the context menu, which creates the link to the selected components. If everything has been linked correctly, the "Execute" button can be clicked in the *fourth phase (4)* in the upper right corner of the window. Thereafter, the result window will be displayed and it will contain a tab for each selected component type that produces a result/report (see Figure 5).

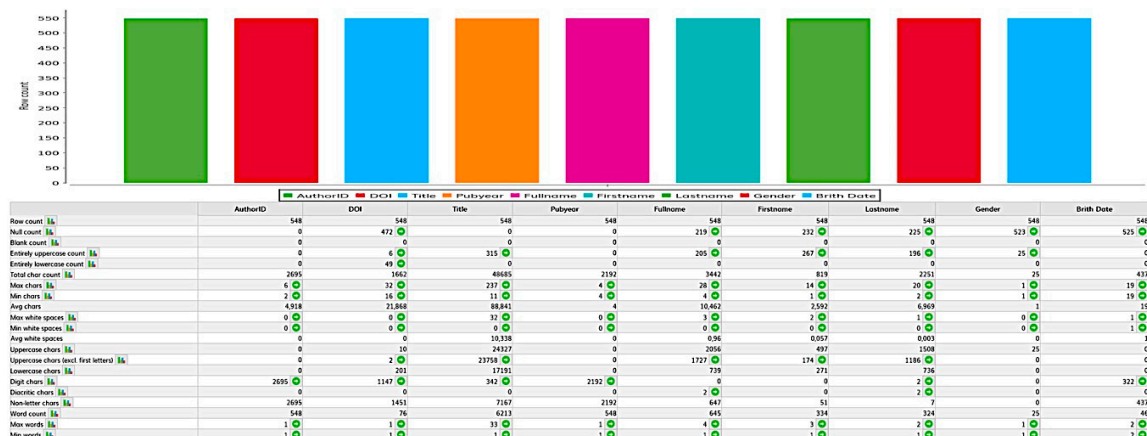

**Figure 5.** Example of a generated result/report from DataCleaner tool.

The considered example provides an insight into the potential for improvement, which can be tapped through different positions of transformations of the data sources into the RIS. The more complex the transformations of the data, the higher the achievable advantages. With DataCleaner, expert users in universities and research institutes can easily and quickly create and use complex data quality rules. In addition to transforming research information to ensure data quality, DataCleaner also offers a variety of techniques (e.g., data profiling and data cleansing) for use, covering all key areas of data quality.

## 5. Conclusions

The success of a RIS depends to a high extent on the quality of the data, which are determined by the extraction and transformation components. ETL process is the consistent development to be able to design complex modeling and analysis processes simply and transparently. The investigations clearly showed that during the transformation phase of the ETL process the processing of the internal and external data sources takes place. In the IT departments, the topic of integrating research information into the RIS traditionally plays a very important role. Many institutions are wondering how they can capture data from multiple data sources and in different formats and move it into single or multiple data stores. As well as summarizing data from different IT systems with different data structures presents a big challenge. To meet these challenges, the goal is to implement the ETL process. It enables the filtering, aggregation, harmonization, linking, cleansing and validation of data that has already been consolidated in the RIS in order to generate new information that may be of special importance to an institution. In the implementation of ETL process there are many of commercial ETL tools in the open source environment (such as CloverETL, etc.) that help integrate the heterogeneous data source system landscape of research information into the RIS.

**Author Contributions:** O.A. contributed the design and implementation of the research, the analysis of the results and wrote the manuscript with input from all authors. Both G.S. and M.A. authors discussed the results of the manuscript and supervised the research project.

**Funding:** This research received no external funding.

**Conflicts of Interest:** The authors declare no conflicts of interest.

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
