# Peer review of "ETL Best Practices for Data Quality Checks in RIS Databases"

_informatics, doi:10.3390/informatics6010010_

Reviewer 1 Report

This paper presents an overview of quality control in Research Information System during extract, transform, and load process. 

Integrating data from multiple sources should consider and closely related to data quality issues, data quality control, and data quality assurance. Many researchers explore and try to work on optimization in extracting, transforming and loading data into valuable integrated data and information. This topic is challenging. By reading the abstract, this paper discusses how the ETL processes conducted during data integration in the case study of commercial research information systems, and hopefully, data quality issues addressed in comprehensive ways.

The methodology and quality criteria model for measuring the quality are well-explained. The introduction and literature review are good in providing the background, but they must be improved (by adding other references in data quality control (QC), quality assurance (QA), and the use of ETL from other perspectives or in the same research domain).

Page 1 Line 40, the definition of RIS (or Current RIS?) and type of RIS can be added with other references. What is the difference between a database and a federated information system? The terms of "data" and "information" are interchangeable, but there is a distinct.

Page 2 Line 28, the definition of data quality is well-described, I am looking for the definition of quality control and quality assurance. My suggestions are adding references and explanation on how quality assurance conducted in the research. Is there any specific document as part of quality assurance activities? My other suggestion is removing keyword: "research information quality assurance"

The use of "research information quality perception" brings me to bigger expectation on how the implementation of control quality with the help of ETL was evaluated. Unfortunately, I cannot find it. It would be great if the research put this information that generated from "user" perception (for example using a questionnaire of perceived quality). My other suggestion is removing keyword: "research information quality perception"

Page 7 Sub-sections of Section 4. I would suggest using the quality criteria model (4.1 Completeness, 4.2. Correctness, 4.3. Consistency and 4.4 Timeliness) as the title of each sub-section.

Page 11 Line 1, the quality of figure 6 need to be improved and add some text (annotation) of the three-part in the figure.

Overall the paper is interesting to read. However, it can be improved and I believe this is a very useful contribution to the informatics field in general, and to the Research Information System domain in specific.

Author Response

We thank the first reviewer for taking the time and carefully reading our work and helpful comments on improving the manuscript. We have made the recommended corrections below and hope that it will meet your expectations. Our comments are written in green.

The methodology and quality criteria model for measuring the quality are well-explained. The introduction and literature review are good in providing the background, but they must be improved (by adding other references in data quality control (QC), quality assurance (QA), and the use of ETL from other perspectives or in the same research domain).

We have edited the other references from other perspectives of ETL in the text and added them to references. Unfortunately, there are no references for this same research domain (e.g., ETL for RIS).

Page 1 Line 40, the definition of RIS (or Current RIS?) and type of RIS can be added with other references. What is the difference between a database and a federated information system? The terms of "data" and "information" are interchangeable, but there is a distinct.

CRIS or RIS is the same, we have made it clearer in the introduction with other references.

I will define the two terms more distinctly, so that the difference becomes clear:

A database or database system is a system for electronic data management. This essential task is to store large volumes of data efficiently, without contradictions and permanently and to provide necessary subsets in different, needs-based forms of presentation for users.

A federated information system or often called a federated database system is an information system that provides access to multiple autonomous sources of information without copying their data, such as in a data warehouse. The form of integration of multiple data sources, where they themselves remain unchanged, is also referred to as virtual information integration. "Federated" is the name of this system because it is a combination of individual systems that maintain their independence.

In this case, RIS can be as a central database or federated information system like Data Warehouse.

Page 2 Line 28, the definition of data quality is well-described, I am looking for the definition of quality control and quality assurance. My suggestions are adding references and explanation on how quality assurance conducted in the research. Is there any specific document as part of quality assurance activities? My other suggestion is removing keyword: "research information quality assurance" We have edited your suggestion and deleted the two terms (QC and QA) in the manuscript to avoid confusion.

The use of "research information quality perception" brings me to bigger expectation on how the implementation of control quality with the help of ETL was evaluated. Unfortunately, I cannot find it. It would be great if the research put this information that generated from "user" perception (for example using a questionnaire of perceived quality). My other suggestion is removing keyword: "research information quality perception". We have deleted the term research information quality perception in keywords.

Page 7 Sub-sections of Section 4. I would suggest using the quality criteria model (4.1 Completeness, 4.2. Correctness, 4.3. Consistency and 4.4 Timeliness) as the title of each sub-section.

We would like to take your suggestion, but subsection of section 4 deals with the steps of the transformation phase in ETL, where I worked out the examples using publication data from Web of Science in DataCleaner tool. The data quality dimensions must always be considered in the process of transforming data, but we have examined these in another manuscript.

Page 11 Line 1, the quality of figure 6 need to be improved and add some text (annotation) of the three-part in the figure.

We have improved the quality of Figure 7 again and added the explanation of the practical example in the developed tool step by step. At the end a result example is attached.

We hope, Figures 7 and 8 are better for reading and understanding.

Reviewer 2 Report

Summary

This paper presents an overview of quality control methods for the integration of information in Research Information Systems. The paper first provides the basic concepts of data quality in traditional ETL integration settings and then it presents best practices of data normalization \ transformation techniques during the integration process.

General Comments

The article covers the well-known area of data quality in the integration of information from different data sources. Its contribution however is limited; it states that it addresses RIS cases, although it is not clear what are the specific requirements posed by such Information Systems. The whole presentation and all cases of data quality metrics are generic and applicable to almost all types of IS  and ETL processes. A specialization on the requirements posed from RIs is needed. For example

A) A more concrete example , in the introduction, on the type of information processed and integrated in RIS is needed . E.g., publications? Books? Management of research resources?

B) In section 2, please consider an extra cause of many data inconsistencies (affecting the overall data quality) that is due to the evolving nature of the data sources, i.e., their schema and semantics evolve over time. See related work for quality metrics for ETL ecosystems evolution from Vassiliadis, Papastefanatos et al. It is worth mentioning this aspect as well.

C) Section 4 present common normalization techniques for data cleaning and improvement. However, there is plenty of research works in the area of data integration regarding name and entity matching.  I think that information integration in RIS greatly involves tasks for entity matching, like authors deduplication, publication matching, and enrichment. These task do not always follow exact matching \ transformation techniques but also involve approximate solution due to errors imposed by data entry, by heterogeneity in schemas\semantics in the data sources, etc. I would propose to mention such techniques in the proposed solutions. See for example the article from P.Christen - ”A Comparison of Personal Name Matching: Techniques and Practical Issues” or the most recent works on blocking methods used in current IS for entity resolution, see e.g., Papadakis et al ”Schema-agnostic vs schema-based configurations for blocking methods on homogeneous data”

D) Figure 6 is not readable and it does not help the reader understand its relevance to the overall paper. Is it a tool you have developed that is used in RIS integration? If so, please consider explaining in a few more details on its practical use.

E) Have you any real-world example or real-world results that can be presented before the conclusions, on the insights gained from the application of these techniques? It would definitely help the reader understand what is the value added by improving these metrics.

Author Response

We thank you for taking the time to review our manuscript. We have taken note of your comments or suggestions and observed them during processing. Thank you again and hope we could meet your expectations.

A) A more concrete example , in the introduction, on the type of information processed and integrated in RIS is needed . E.g., publications? Books? Management of research resources?

We have created a Figure (see Fig.1) and shown clearly which research information can be integrated into the RIS.

B) In section 2, please consider an extra cause of many data inconsistencies (affecting the overall data quality) that is due to the evolving nature of the data sources, i.e., their schema and semantics evolve over time. See related work for quality metrics for ETL ecosystems evolution from Vassiliadis, Papastefanatos et al. It is worth mentioning this aspect as well.

Thank you for the interesting paper, I have read it and added an inconsistent example of the data in the manuscript (see section 2), with the literature you suggested.

C) Section 4 present common normalization techniques for data cleaning and improvement. However, there is plenty of research works in the area of data integration regarding name and entity matching.  I think that information integration in RIS greatly involves tasks for entity matching, like authors deduplication, publication matching, and enrichment. These task do not always follow exact matching \ transformation techniques but also involve approximate solution due to errors imposed by data entry, by heterogeneity in schemas\semantics in the data sources, etc. I would propose to mention such techniques in the proposed solutions. See for example the article from P.Christen - ”A Comparison of Personal Name Matching: Techniques and Practical Issues” or the most recent works on blocking methods used in current IS for entity resolution, see e.g., Papadakis et al ”Schema-agnostic vs schema-based configurations for blocking methods on homogeneous data”

Thank you for your suggestion, these methods e.g. schema matching and schema mapping I examined in another published paper (see Literatur [5]), in the context of two standards CERIF from the organization EuroCRIS and German KDSF from our institute.

We now want to focus on the transformation of the data because many mistakes are made in this process. We try to explain and solve the data quality problems of the research information during the transformation phase using ETL. But we clearly mentioned that at the beginning of Section 4. This manuscript can be considered as a sequel to the other published paper [5]. Your suggested literature from P.Christen and Papadakis, Alexiou, Papastefanatos and Koutrika are cited in the other published paper [5].

D) Figure 6 is not readable and it does not help the reader understand its relevance to the overall paper. Is it a tool you have developed that is used in RIS integration? If so, please consider explaining in a few more details on its practical use. We have improved Figure 7 and explained the practical application with the tool step by step to make it clearer to the reader.
The tool was developed by a company but we have expanded it at the institute and I use it to analyze and improve the research information before its integration into the RIS.
We hope, Figures 7 is better for reading and understanding.

E) Have you any real-world example or real-world results that can be presented before the conclusions, on the insights gained from the application of these techniques? It would definitely help the reader understand what is the value added by improving these metrics. We have added a result example (see Fig.8) before the conclusion and it refers to Figure 7.

We thank you again for your careful reading and your helpful comments on improving our manuscript!

Round  2

Reviewer 1 Report

I am satisfied with the response. I thank the authors for accepting my suggestions. Thanks a lot for all these improvements.

The new manuscript title "ETL Best Practices for Data Quality Checks in RIS 3 Databases" is clear, attractive and describe the study. Furthermore, It is supported by a clear introduction, a well-explained methodology, and good result.

The explanation of the practical example in the developed tool step by step make the research more easily understandable for the readers. Moreover, additional references have been included in this manuscript version.

I am satisfied because the manuscript has been significantly improved. Therefore I accept the paper in its current form.